## REVIEW ARTICLE

# Computational methods to simulate molten salt thermophysical properties

Talmage Porter[1,2], Michael M. Vaka[1,2], Parker Steenblik[1] &
Dennis Della Corte [1 ✉]

Molten salts are important thermal conductors used in molten salt reactors and solar applications. To use molten salts safely, accurate knowledge of their thermophysical properties is necessary. However, it is experimentally challenging to measure these properties and a comprehensive evaluation of the full chemical space is unfeasible. Computational methods provide an alternative route to access these properties. Here, we summarize the developments in methods over the last 70 years and cluster them into three relevant eras. We review the main advances and limitations of each era and conclude with an optimistic perspective for the next decade, which will likely be dominated by emerging machine learning techniques. This article is aimed to help researchers in peripheral scientific domains understand the current challenges of molten salt simulation and identify opportunities to contribute.

**General introduction to molten salts**. Molten salts are ionic mixtures that are solid at standard temperature and pressure and liquid at elevated temperatures. Molten salts have a high heat capacity and good thermal conductivity which makes them useful as heat exchangers and thermal capacitors in applications such as molten salt reactors and next generation solar systems[1]. Molten salts include simple systems such as alkali-halides, oxides, and various other salts. For practical applications, complex salts, such as LiF-LiBe2 (Flibe), are of greater interest; opening the door to a large set of salt mixtures to be explored. Extracting thermophysical properties from molten salts experimentally is challenging due to their toxicity, high temperature, and cost of doing so. This motivates the accurate characterization of molten salts through computational methods.

Initial interest in molten salt simulations is associated with the molten salt reactor experiments conducted at the Oak Ridge National Laboratories during 1965–1969. Molten salt reactors are nuclear fission reactors in which the primary coolant and/or fuel is a molten salt mixture. Molten salt reactors are considered promising candidates for Generation IV reactor designs which have sparked new interest and investigation into computational methods to improve understanding of molten salt properties[2,3]. A visualization of a molten salt simulation box is shown in Fig. 1.

Throughout the last 70 years, various breakthroughs in molten salt simulations can be traced back to novel theoretical approaches, experimental techniques, materials, and increased computational power[2,3]. Within this review, we focus on three principal epochs: *Early simulations* from 1930–1990, *Progression of DFT-based methods* from 1990–2021, and the *Machine Learning Era* starting in 2020. For each period, we review the new technologies developed followed by their achievements and limitations. During a time when Machine Learning is disrupting many industries and research fields, this review can serve as an introduction for tangentially interested research groups not yet familiar with the potential impact of improved simulation techniques for molten salts.

[1] Department of Physics and Astronomy, Brigham Young University, Provo, UT, USA. [2] These authors contributed equally: Talmage Porter, Michael M. Vaka.
✉email: Dennis.DellaCorte@byu.edu

**Introduction to thermophysical properties.** The aim of computational simulations of molten salts is the accurate prediction of thermophysical properties. These properties are required to simulate complex fluid dynamics and chemical interactions within molten mixtures. Highly accurate and fast computational explorations of different molten salts could replace expensive and hazardous experiments. The properties of interest are categorized into two classes: (1) Properties that can be derived from short simulations of small systems with only a few explicitly modeled atoms, like heat capacity, density, etc.; (2) Properties that require long simulation times and large system sizes before converging to experimental accuracies, like diffusion, thermal conductivity, etc. Figure 2 summarizes reported properties from reviewed articles and the simulations sizes and durations used to calculate them. Modeling the second class of properties has driven the exploration of more efficient simulation techniques, albeit often at the cost of accuracy.

The derivation of thermodynamic properties for a molten salt requires the atomistic simulation of a system consisting of a specified number (N) of atoms from each chemical component (for example Flibe has been simulated with 100 F, 200 Li, and 400 Be atoms). Various methods exist for simulating the movement of atoms in these systems. Common elements of most simulations are: (1) Each atom is modeled as a point in space with a given

position, velocity, and charge attributes; (2) The interaction between atoms is calculated to yield a potential energy term; (3) Forces per atom are calculated as the negative gradient of the potential energy term at a given position; (4) An integrator is used to solve Newton's equations of motion for each atom and to update the positions. Executing such simulations under defined constraints—such as constant pressure (P), temperature (T), energy (E), or volume (V)—results in dynamic trajectories of the system. A trajectory is an ordered collection of frames, where each frame represents a snapshot in time of the atomic positions of the simulated system. Figure 3 shows a decision flow chart that can be used to select the right simulation for a desired thermophysical property.

The theories of statistical mechanics provide the recipes to derive thermophysical properties from such simulation trajectories. Structural properties are far more accurate to obtain experimentally. These properties are experimentally derived through neutron scattering experiments and Extended X-ray Absorption Fine Structure. The most common derived properties for MD are the partial radial distribution function, coordination number, angular distribution function, bond angle, and the structure factor. A comprehensive summary of all thermophysical and structural properties is given below.

## Overview of the mathematics behind thermophysical properties

*Temperature.*

$$T = \frac{1}{3Nk_B}\sum_i^N m_i v_i, \tag{1}$$

where $N$ is the number of atoms, $m$ is the mass, and $k_B$ is Boltzmann's constant. Velocities, $v_i$, can be calculated for each particle by taking the change in their position between time trajectory frames.

*Density.*

$$\rho = \frac{NM}{N_\alpha V}, \tag{2}$$

where $N$ is number of atoms, $M$ is molar mass, $N_\alpha$ is Avogadro's number, and $V$ is the equilibrated volume of the simulation cell of the given temperature in the isothermal-isobaric (NPT) ensemble. An alternative to determining the density through the NPT ensemble is to use previously measured experimental values for the density. Using experimental densities in some cases is more

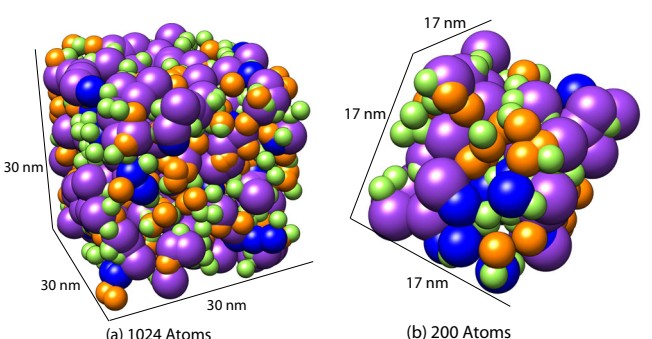

**Fig. 1 Comparison of simulation sizes used in Machine Learning and DFT calculations.** Each ion type is represented by a color—Fluorine in green, Sodium in blue, Potassium in pink, and Lithium in orange. **a** Simulation size used with a Machine Learning Potential with a total simulation size of 1024 atoms and a side length of 30 nm. **b** Simulation size used with DFT calculations with a total simulation size of 200 atoms and a side length of 17 nm.

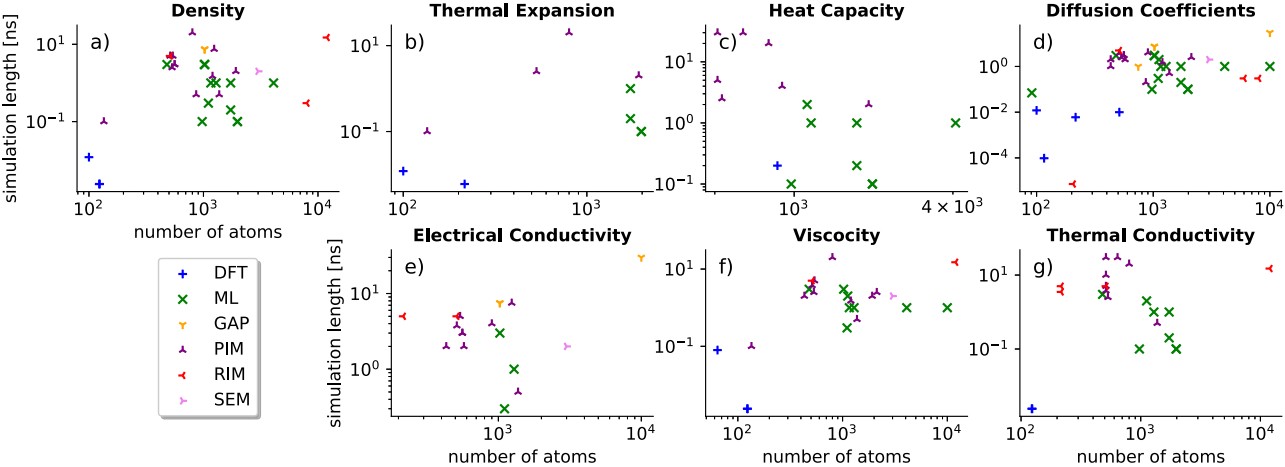

**Fig. 2 Thermophysical properties extracted from reviewed articles.** Colors and symbols differentiate between computational methods used in the studies (see legend in left bottom panel). For seven thermophysical properties (**a**–**g**), the number of atoms and the total simulation lengths are plotted.

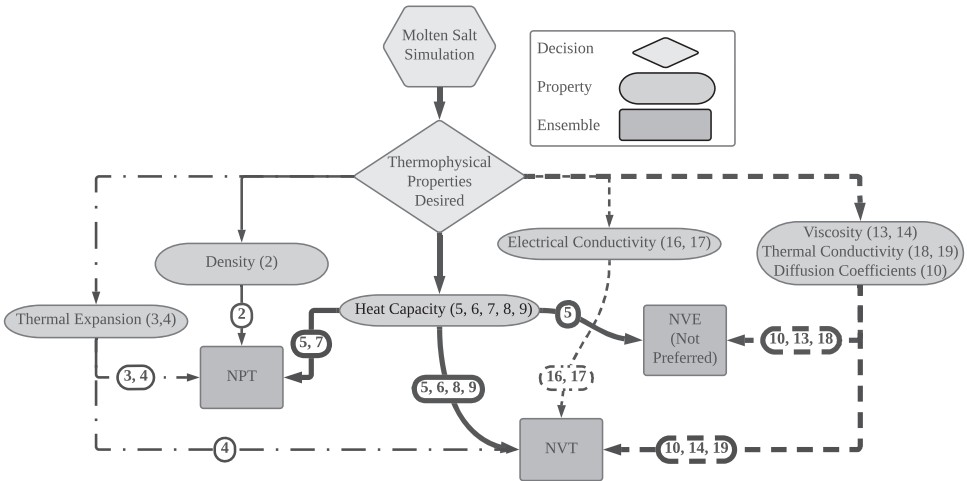

**Fig. 3 Simulation strategies for molten salts to obtain thermophysical properties.** Each property is followed by the equation number listed in the text that can be used to derive the property from the simulation data. The arrows connecting the properties with the simulation ensemble (NPT, NVE, and NVT) are again marked with the equation numbers. For example, thermal expansion can be calculated from NVE simulation with Eq. 3 or from NPT simulation with Eq. 2, however, the NPT simulation is the preferred route. Reverse non-equilibrium molecular dynamics (RNEMD) and equilibrium molecular dynamic (EMD) are explained further in the text.

accurate in the calculation of the other properties such as diffusion, viscosity, thermal, or electrical conductivity. In a nuclear system, high-density molten salts increase neutron production which causes the system to approach criticality. However, if the density is too low the salt will only sustain the system for a short period. An alternative method to find the equilibration volume is to run canonical ensembles (NVT) at different volumes and then fit the pressures and volumes with a Murnaghan equation of state to find the equilibration volume and bulk modulus[4].

*Thermal expansion.*

$$\beta = \frac{1}{\rho}\left(\frac{\partial \rho}{\partial T}\right)_P, \tag{3}$$

$$\beta = \frac{1}{V}\left(\frac{\partial V}{\partial T}\right)_P, \tag{4}$$

Equation (3) is more commonly used than (4). The results of the NPT ensemble can be used to fit the relation between density and temperature. The relation between density and temperature can be used to find the thermal expansion at a specific density using Eq. (3). Equation (4) can be used by running many NVT ensembles and then using the Murnaghan equation of state to find the equilibration volume for different temperatures[5]. Fitting a line to the relation between the equilibration volumes and temperatures yields the thermal expansion at a specific volume.

*Heat capacity.*

$$C_P = \left(\frac{\partial H}{\partial T}\right)_P = \left(\frac{\Delta(U + PV)}{\Delta T}\right)_P, \tag{5}$$

where $C_P$ is the heat capacity at constant pressure, $H$ is the specific enthalpy, $T$ is temperature, and U is the system's total internal energy including both kinetic and potential energy. This can be simulated in NPT or the NVT for a range of target temperatures.

$$C_V = \left(\frac{\partial U}{\partial T}\right)_V, \tag{6}$$

Equation (6) is the heat capacity at a constant volume. For MS $C_V$ is usually close to $C_P$ and has very little temperature

dependence.

$$C_P = \frac{\langle \delta E^2 \rangle_{NPT}}{k_B T^2}, \tag{7}$$

$$C_V = \frac{\langle \delta E^2 \rangle_{NVT}}{k_B T^2}, \tag{8}$$

are other approximations commonly used, where $\langle \delta E^2 \rangle = \langle E^2 \rangle - \langle E \rangle^2$ and $\langle E \rangle$ denotes the average. $T$ is the temperature and $k_B$ is the Boltzmann constant.

Equation (9) is another approximation similar to Eq. (7):

$$C_V = \frac{\Delta E_{NVT}}{k_B T^2}, \tag{9}$$

where $\Delta E$ is the fluctuation in energy[6].

*Diffusion coefficients.*

$$D_\alpha = \frac{1}{6t}\lim_{t \to \infty}\left\langle \left| \delta \mathbf{r}_i(t) \right|^2 \right\rangle, \tag{10}$$

where $\langle |\delta r_i(t)|^2 \rangle$ is the mean square displacement of the element α, meaning: square the displacements $\delta r$ of each particle $i$ at various times $t$ and then take the average. The six in the denominator results from being 3-dimensional diffusion. After calculating diffusion coefficients, it is common to find the Arrhenius relationship for temperature and diffusion coefficients:

$$D(T) = D_\alpha e^{-\frac{E_\alpha}{R}T}, \tag{11}$$

where $E_\alpha$ is the activation energy of diffusion, $R$ is the ideal gas constant, and $T$ is the temperature. This relationship is used to find the diffusion activation energy. Alternatively, the solubility is defined by:

$$S(T) = S_\alpha e^{-\frac{H_\alpha}{R}T}, \tag{12}$$

where $H_\alpha$ is the enthalpy of mixing and $S_\alpha$ is the solubility as the temperature goes to infinity.

*Viscosity.*

$$\eta = \frac{V}{k_B T}\int_0^\infty \langle \sigma_{\alpha\beta}(t) \cdot \sigma_{\alpha\beta}(0) \rangle dt, \tag{13}$$

uses the Green-Kubo relation through the integration of the shear stress autocorrelation function under an NVT ensemble where $\sigma$ is the virial pressure tensor. It is averaged with respect to the off-diagonal components (i.e., $\alpha \neq \beta$).

$$\eta = -\frac{\sum\limits_{\text{transfer}} m(v_{\text{xhot}} - v_{\text{xcold}})}{2tL_xL_y\left\langle\frac{\partial v_x}{\partial z}\right\rangle}, \tag{14}$$

can be used to calculate viscosity using a reverse non-equilibrium molecular dynamics (RNEMD) method in the microcanonical (NVE) ensemble. Where $L$ is the size of the simulation box, $T$ is the temperature, $v$ is the velocity of the particles, $m$ is their mass, and $x$, $y$, $z$ are the coordinates. $\eta = \frac{k_BT}{2\pi D\lambda}$, is a third option that is more of an approximation than the others. Where $\lambda$ is the step length of ion diffusion usually assumed to be the diameter of the ion.

Viscosity also follows the Arrhenius relationship in Eq. (11) just like diffusion. And the activation energies of the viscosity can be found from it:

$$\eta(T) = \eta_\alpha e^{-\frac{E_a}{R}T}. \tag{15}$$

*Electrical conductivity.*

$$\sigma = \frac{1}{3Vk_BT}\int_0^\infty \langle J_z(0) \cdot J_z(t)\rangle dt, \tag{16}$$

is the most frequently used method using the Green-Kubo relation and the autocorrelation function where $J_z(t) = \sum_{i=1}^N z_i e v_i(t)$. Here $z_ie$ is the charge of the particle $i$, $V$ is volume, $k_B$ is the Boltzmann constant, and $T$ is temperature.

$$\sigma = \frac{e^2}{6tk_BVT}\lim_{t\to\infty}\left\langle\left|\sum_\alpha z_\alpha e\Delta\alpha(t)\right|^2\right\rangle, \tag{17}$$

resembles the diffusion equation and employs the mean squared displacement (MSD) of the particle $\alpha$. Notice this is an EMD method using the NVT ensemble. Equation (16) using the autocorrelation function is considered more accurate than using the MSD.

A third uncommon approximation can be made with: $\sigma = D\frac{nZ^2e^2}{k_BT}$.

*Thermal conductivity.*

$$\lambda(T) = \frac{V}{3k_BT^2}\int_0^\infty \langle J_E(t) \cdot J_E(0)\rangle dt, \tag{18}$$

uses the Green-Kubo relation through the integration of the auto-correlation of $J_E$, simulated in the NVT ensemble. Where $J_E = \frac{1}{V}\left[\sum_{i=1}^N E_i v_i + \frac{1}{2}\sum_{j\neq i}^N (r_{ij}f_{ij})v_i\right]$ and $E_i = \frac{1}{2}[m_iv_i^2 + \sum_{j\neq i}^N U_{ij}(r_{ij})]$ is a summation of kinetic and potential energies. Where $r_{ij}f_{ij}$ is the position and force on a particle, $V$ is the volume, $v$ is the velocity, $m$ is the mass, $k_B$ is Boltzmann's constant and $T$ is the temperature.

There is also a useful RNEMD method for calculating the thermal conductivity in NVE similar to Eq. (14):

$$\lambda = -\frac{\sum\limits_{\text{transfer}} \frac{m}{2}(v_{\text{hot}}^2 - v_{\text{cold}}^2)}{2tL_xL_y\left\langle\frac{\delta T}{\delta z}\right\rangle}, \tag{19}$$

where $L$ is the size of the simulation box, $T$ is the temperature, $v$ is the velocity of the particles, $m$ is their mass, and $x$, $y$, $z$ are the coordinates.

*Partial radial distribution function (PRDF).* PRDF, sometimes referred to as RDF, is defined[7] as:

$$g_{\alpha\beta}(r) = \frac{1}{4\pi\rho_\beta r^2}\left[\frac{dN_{\alpha\beta}(r)}{dr}\right], \tag{20}$$

or alternatively[8] as

$$g_{\alpha\beta}(r) = \frac{1}{\langle\rho_\beta\rangle}\sum_{i\in\alpha}^{N_\alpha}\sum_{j\in\beta}^{N_\beta}\frac{\delta(r_{ij} - r)}{4\pi r^2}, \tag{21}$$

where $\rho_\beta$ is the number density of species $\beta$, $N_\alpha$ is the number of the respective species $\alpha$, and $N_{\alpha\beta}(r)$ is the mean number of $\beta$ ions lying in a sphere of a radius $r$ centered on an $\alpha$ ion. This function can be thought of as measuring the correlation between two atoms.

*Coordination number (CN).* The CN is defined as:

$$N_{\alpha\beta} = \rho_\beta\int_0^{r_{\min}} g_{\alpha\beta}(r)4\pi r^2dr, \tag{22}$$

where $r_{\min}$ the position of the first valley in the PRDF.

*Angular distribution function.* The angular distribution function can be thought of as the correlation between three particle types and can be derived from the bond angle shown below.

*Bond angle.* The bond angle is defined as:

$$\theta_{ijk} = \left\langle\cos^{-1}\left(\frac{r_{ij}^2 + r_{ik}^2 - r_{jk}^2}{2r_{ij}r_{ik}}\right)\right\rangle, \tag{23}$$

where $i,j,k$ are the central atom and two arbitrary atoms indices, respectively. This function can be thought to encode the atomic configuration types and the correlation between three-particle types[7].

*Structure factor (S).* The structure factor is defined as the Fourier transform of the PRDF in the following manner:

$$S_{ij}(k) = \delta_{ij} + 4\pi\sqrt{\rho_i\rho_j}\int_0^\infty\left[g_{ij}(r) - 1\right]\frac{\sin(kr)}{kr}r^2dr, \tag{24}$$

where $\rho_\alpha$ is the number density of the $\alpha$ ions[9].

**Outline of this manuscript.** This review presents the evolution of computational simulations in three eras, culminating in the combination of molecular dynamic simulations with machine learned potentials that are trained on ab-initio data. This new breed of simulation techniques enables the accurate simulation of large molten salt systems for extended periods of time, sufficient to predict thermophysical properties otherwise not accessible. Current techniques, however, rely on outdated neural network approaches that are insufficient to capture the interactions between complex molten salt mixtures. We present the strengths and weaknesses of the different methods and provide insights where new methods could overcome the existing challenges.

## Method

We followed a systematic literature review process to gather the relevant publications for this review. The keywords *ionic liquid*, *molten salt*, and *molten salt reactor* were used in the advanced search of Google scholar to identify a list of 138 potentially relevant papers. After reading the abstracts, the papers were sorted into potentially relevant and not-relevant papers. Each potentially relevant paper was read in detail and summarized by the authors. This process resulted in 57 papers found to be of relevance. In a second step, the cited references of each relevant

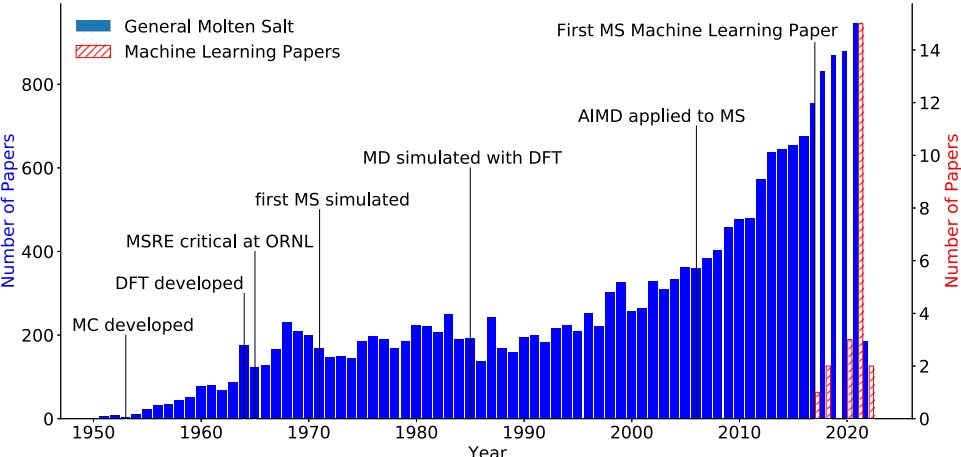

**Fig. 4 Timeline of papers published on molten salt reactors per year.** Generally relevant papers in blue. Papers that use machine learning and apply it to molten salts are shown proportionately in hatched-red. Important milestones are marked and include: Monte Carlo (MC) being developed, Density Functional Theory (DFT) being developed, the Molten Salt Reactor Experiment (MSRE) going critical at Oak Ridge National Labs, the first molten salt (MS) simulation, Molecular dynamics (MD) begin to be simulated with Density Functional Theory, Ab Initio Molecular Dynamics (AIMD) are applied to molten salts, and machine learning being applied to Molten Salts.

paper and the citing references were collected and checked for relevance. At the end of this process, 95 total papers were identified. An additional search for machine learning papers was conducted, which added 24 papers to the total relevant number of papers for this review. Figure 4 shows a breakdown of related publications per year and is indicative of the wave-like interest in molten salts throughout the different eras.

## Discussion

**Early simulations 1933–1990**. The development of quantum mechanics and the Born-Oppenheimer approximation led to the first potentials for alkali halides that were fitted semi-empirically in the condensed phase. Starting in 1933, computationally derived properties were being reported alongside the development of the Born-Huggins-Mayer (BHM) model[10]. However, limited by the potential form and the lack of computational power, the first set of estimated parameters was rather approximate to experiment[11].

The development of the Monte Carlo method in 1953 opened new simulation possibilities and Tessman et al. found polarizabilities of the alkali halides, which were essential for more accurate simulations[12]. New ionic salt models were developed to improve the accuracy of derived behavior. In 1958, Dick and Overhauser developed the shell model as the first attempt to capture induced polarization effects in ions[13]. By 1959, the Molecular Dynamics method was introduced which enabled the simulation of trajectories for molten salts, and consequentially the development of interatomic potentials continued to grow[14]. Tusi and Fumi modified the Born-Mayer-Huggins potential (BHMTF) and some parameters in 1964 using semi-empirical methods and improved predicted properties, such as densities, diffusion, and heat capacity[11,15]. Besides the still limited computational resources, many of the developed potentials suffered from inaccuracies in fitted parameters—such as Vander-Waals coefficients—, lack of treatment of many-body interactions, and, especially for the shell model, arbitrary assumptions made about potential forms, all of which left much room for improvement[5].

A breakthrough for atomistic simulation was the development of density functional theory (DFT) by Hohenberg and Kohn in 1964[16]. With the ability to calculate accurate potentials, ab-initio calculations became possible and opened the door to computational—albeit not very feasible—first-principle simulations.

During the same time, molten salt experiments climaxed with the execution of the Oak Ridge Molten Salt Reactor Project between 1965 and 1969. Consequent publications in 1970 described the chemical considerations that motivated the choice of specific salts in Oak Ridge; such salts being of the alkali halide family as well as others[17,18].

The first simulation of molten salts used KCl and was reported in 1971 by Wood, Cock, and Singer using Monte Carlo with the BHMTF potential[19]. Others continued through 1976 developing various rigid and polarizable models to investigate alkali halides[11]. From these investigations, it became apparent that many-body interactions would need to be included in some explicit manner. In 1983, Tang and Toennie introduced a universal dispersion damping function that could be applied to Born-Mayer type potentials which were one of the first attempts towards the inclusion of many-body effects[20]. In 1985, Madden and Fowler expanded this idea using their asymptotic model of polarization, expanded terms beyond the dipole-induced dipole model, which was comparable to LiF simulations as obtained from Hartree-Fock calculations[21]. It was also during 1985 that ab-initio MD (CPMD), by Car and Parrinello, was introduced; it coupled molecular dynamics with a DFT potential[22]. The tools of this era laid the framework upon which the next would be built.

**Progression of DFT-based methods 1990–2021**. Increased computational power and theoretical insights advanced interatomic potentials and density functional theory to the level of usefulness and reliability. Both fields were developed by different research groups whose insights influenced each other. Here, we will first review the progression of DFT-based methods, resulting in dispersion corrected ab-initio molecular dynamics (AIMD) methods. Afterward, we will discuss the advancements of interatomic potentials, as they were strongly influenced by the increase in accuracy and feasibility of DFT models.

In 1993, Barrnet and Lanmarn introduced Born-Oppenheimer molecular dynamics with DFT which enabled larger simulation timesteps resulting in a better temporal sampling of still very restricted simulations[23]. Later that year, Kresse and Hafner introduced the method of initializing Car-Parrinello Molecular Dynamics with energy minimization schemes for metals, resulting in additional speed up[24]. In 1998, Alfe extracted the diffusion coefficients using CPMD for liquid aluminum and

thereby demonstrated the feasibility of DFT methods to extract transport properties in condensed materials[25]. In 2003, Aguado et al. developed an ab-initio process using many condensed phases of MgO with CATSTEP DFT to parametrize coefficients for the aspherical ion model (AIM) representing a breakthrough for polarizable models[26]. In 2005, Hazebroucq used a tight-binding density functional to calculate and investigate diffusion in NaCl and KCl for a specific experimental volume, a step towards full AIMD[27].

In 2006 Madden et al. highlighted the need to control the dispersion interaction as it—despite being only a tiny fraction of the interaction energies of an ion pair—strongly impacted phase transition behavior, such as transition pressures[28]. It had been a well-known problem up to this time that dispersion interactions in DFT calculations were not accurate. Grimme tackled the problem of dispersion and published a first empirical correction in 2006 for DFT followed by a second in 2010, practically resolving the issue[29]. In 2006, Klix investigated the diffusion of tritium in Flibe, using CPMD, and named the method ab-initio molecular dynamics[30]. This simulation was the first time that a molten salt had dynamical behavior derived using ab-initio methods.

In 2014, Corradini investigated dispersion in LiF with both DFT and molecular dynamics and found that dispersion is significant in NPT simulations: it strongly affected melting point calculations and resulted in underestimated equilibrium densities by 15% when omitted and thereby verified the importance of dispersion in molten salt calculations[31]. In 2015, Anderson used AIMD to model FLiNaK and Flibe and extracted thermodynamic properties such as density, diffusion, and thermal expansion which were validated by experimental measurements[32]. From 2016–2021 multiple papers investigated the thermophysical behaviors for various salts using AIMD[3,33–43]. Few of the papers investigated thermodynamical quantities and even less investigated kinetic properties (diffusion and viscosity)[34,37,42]. Short timescales and a small number of ions render such calculated properties unreliable which most, if not all, DFT-based calculations had in common.

Optical properties such as vibrational spectra are more accessible experimentally than computationally and only a small number of DFT studies investigated them[33,44]. In 2021, Khagendra et al. reported mechanical properties of Flibe using DFT suggesting that these properties could validate models[44]. However, as these properties are not consistently presented throughout the literature there is currently limited development in this domain.

The development of interatomic potentials for molten salts between 1990 and 2008 converged in the Dipole-induced polarizable ion model (PIM). The basic idea of PIM is that a sufficiently complex additive forcefield can be fitted to DFT data to result in accurate energy estimates. These energies can in turn be used to propagate the atoms in the model forward in time. Accurate DFT models allowed for parametrization of complex forcefields; for example in 2003 the first fitted dipole model was produced[26]. In 2008, Salanne developed PIM and defined the most frequently used interatomic potential for the next decade[45]. Throughout 2008–2020, PIM was used to derive various properties for many salts which were often validated by experiments[1–3,6,7,9,31,32,45–80]. From 2018 until today, a revival of sorts, led to alternative potentials, such as the Sharma−Emerson−Margulis (SEM) Drude oscillator model[81], to be developed.

The quest for the PIM began in 1993 when Madden and Wilson introduced the method of parametrization via CPMD simulation for halides on a rigid ion model (RIM) potential plus a dipole term (see Section "Summary of Potentials" for an overview

of interatomic potentials). This initial model was a response to the earlier shell model; to correct the apparent lack of justification for the potential form[82]. Since oxides could not be accurately described by polarization effects, the Compressible Ion Model was developed in 1996[83]. In the same year, Madden and Wilson enhanced the 1993 model with the addition of a quadrupole using the asymptotic model of polarization and applied it to AgCl systems[84]. Wilson and Madden suggested that ionic salt interactions can be described by four terms: induction/polarization, dispersion, compression, and shape effects[85].

Another stride was taken in 1998 when Rowley and Jemmer published the Aspherical Ion model which combined the Compressible Ion Model with polarization effects from the asymptotic model of induced polarization[25]. This model was used to investigate the polarizabilities and hyperpolarizabilities of LiF, NaF, KF, LiCl, NaCl, KCl, LiBr, MgO, CaO by Jemmer, Madden, Wilson, and Fowler[86].

The importance of polarization was further evidenced in a 2001 study by Hutchinsons, describing trichloride system phase transitions[87]. In 2002, Domene used AIM as a starting point to derive an ion model including dipole and quadrupole moments resulting in another polarizable ion model which they used to investigate properties of MgF2, CaF2, LiF, and NaF[88].

Computationally less demanding interatomic potentials based on the Born-Huggins-Mayer-Tusi-Fumi (BHMTF), or rigid ion model (RIM), also found application in this century. In 2004, Galamba used BHMTF and born-dispersion parameters to calculate the thermal conductivity of NaCl and KCl—although overpredicting it by 10–20%[89]. Brookes used a simpler Born-Mayer potential in 2004 to calculate diffusion and viscosity relationships for KCl, KLi, Sc3Cl, Y3Cl, and La3Cl[90].

In 2006, Madden et al. described the method of first-principle parameterization for molten salts as well as potential forms for polarizable ion models developed up to that point[28]. Heaton et al. were first to simulate Flibe with the BHMTF model and Tang-Teonnie's dispersion damping[91].

By 2007, Galamba investigated the theory of thermal conductivity for Alkali-Halides and derived thermal conductivity of NaCl based on BHMTF including non-equilibrium molecular dynamics, but still overestimated measured properties[92]. In a series of papers from 2008 to 2009, Salanne reported polarizabilities of LiF, NaF, KF, and CsF using DFT condensed phases method, produced the modern PIM, and applied it to LiF-NaF-KF, NaF-ZrF4, LiF-NaF-ZrF4 to derive various thermodynamic properties, including experimentally validated electrical conductivities and diffusion coefficients[45–47].

Others built on the works by Salanne and Madden, such as Merlet et al. who reported in 2010 multiple diffusion and transport coefficients for various LiF-KF compositions[49]. Also in 2010, Olivert reported the structural factors of LiF-ZrF4 from a combination of experiments and PIM-based simulations[50]. In 2011, Salanne and Madden reviewed the importance of polarization effects and promoted the use of PIM[52].

In 2012, Salanne described a method for calculating thermal conductivity using the Green-Kubo formalism and applied it to NaCl, MgO, Mg2SiO4[53]. Another review by Salanne in 2012 compared AIM with PIM and published reusable parameters for alkali halides[54]. In an interesting 2013 study by Benes, heat capacity was determined experimentally for CsF and used to validate DFT and PIM models. While heat capacity was in good agreement with the DFT results, MD simulations did not agree with experimentally determined values for enthalpy of formation[55]. In 2014, Liu investigated Li-Th4, an important MSR salt, using PIM[60].

In 2018, Abramo deployed RIM to model NaCl-KCl liquid-vapor phase, concluding that RIM works well for alkali halides[69].

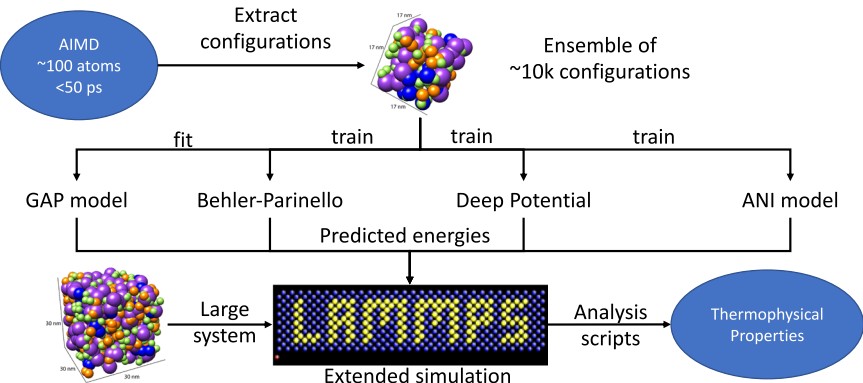

**Fig. 5 Overview of typical methods used in recent machine learning based studies of molten salts.** A short ab-initio molecular dynamics simulation of a small system generates a training set. One of four commonly used models is fitted or trained. A larger system is then simulated using the custom trained potentials for extended periods. The resulting trajectories are analyzed and thermophysical properties are extracted.

In 2019, Guo used PIM to derive thermodynamic properties for Li, Na, K-ThF4[74]. In 2020 Sharma introduced the SEM-drude model which offered an alternative to PIM with 30 times faster execution times[81].

As computational power grew, more accurate DFT-based AIMD simulations became available and spawned sustained interest in the fitting of interatomic potentials with functional forms that could allow rapid evaluation for longer simulation times. The literature up to this point suggests many successful proofs of principle, but a limited number of computational assays that could replace experiments. With Moore's law working in its favor, AIMD might one day become feasible for relevant simulation times and system sizes to derive all thermodynamic potentials, but until then, alternative approaches with higher computational turnaround will remain of high interest.

**Machine Learning Era 2020—ongoing**
*Setting the stage—the promises of machine learning potentials.* Machine learning originated from the work by McCulloch and Pitts, who in 1943 proposed artificial neural networks (ANN) as biologically inspired computation architectures[93]. Many decades passed before useful ANN were constructed, mainly due to increases in computational power and available datasets. Today, old theories are frequently rediscovered and used to design the next breakthrough algorithm, such as ANNs, convolutional neural networks (CNN), or graph neural networks (GNN). Deep learning, a subset of machine learning, serves as a general function approximator that can be trained to mimic any expensive analytical or empirical function, often with substantially reduced execution time. For molten salt simulations, a deep learning method can potentially address the two main problems of existing methods: (1) The limited scalability of ab-initio methods due to their complexity and computational cost for large systems and long timescales; (2) The poor accuracy of efficiently parametrized forcefields due to their limited expressibility.

Novel machine learning tools are frequently evaluated on well-established benchmarks before they find broad adoption. Traditional computer science tasks with well-respected benchmarks, such as data clustering, image annotations, or natural language processing tend to serve as the battleground where new machine learning algorithms prove themselves. Fields like chemical simulations, typically only deploy these models after a knowledge transfer phase. A recent example for this pattern is the attention mechanism[94] that substantially improved natural language processing models in a transformer architecture[95] and has since influenced models like AlphaFold[96].

*Managing expectations—the first applications of neural networks to molten salts.* A review of machine learning models in the field of molten salt simulations is not expected to identify novel machine learning architectures, but rather show the application of well-established methods in a new context. Indeed, before 2020, only one machine learning inspired method was used to predict saturation pressure of pure ionic liquids: Hekayati et al. trained a simple ANN on 325 experimental vapor pressure points and showed that the resulting model could reproduce the experimental values[97]. However, this model was not generalizable and serves only as a historical footnote. The successful applications of machine learning techniques to molten salt simulations began in 2020, using ideas from Behler-Parinello[98] and Bartok[99]. Another related discipline that recently started using ML models is that of nanofluid simulations, however, these typically deal with bulk solvent instead of atomistic simulations and are out of scope of this review[100–107].

The methods based on Behler-Parinello[98] and Bartok[99] have common elements depicted in Fig. 5. Both methods require a set of atomistic configurations and associated energies, typically obtained from ab-initio molecular dynamics of small systems (~100 atoms) and short simulation times (<100 ps). Both are trained to predict the energy of each atom in a configuration so that their sum equals the total energy of the system. The differences and current adoptions follow.

Behler-Parinello proposed a simple feed forward neural network, or multilayer perceptron, that predicts the total energy of a molten salt system as the sum of individual energy contributions of each atom[98]. This idea is like that of the empirical potentials discussed in section "Progression of DFT-based methods 1990–2021". Although, instead of pre-defining a functional form, this approach allows the network to be dynamically optimized during training, potentially considering many-body effects and polarization without explicitly defining them. The main contribution of this work was the definition of a symmetry function that translated the neighborhood of each atom into an input vector for the neural network. For atomistic problems, one of the biggest challenges is to find an appropriate way to represent a collection of atoms so that a trained neural network will obey expected invariances (translation, rotation, atom replacement) of the real world. The original work was only applied to pure silicon systems, so it did not need to deal with different atomistic species.

*Going deeper—extension of basic neural networks for molten salt simulations.* The Behler-Parinello[98] concept was further extended by Han et al.[108] and Zhang et al.[109] in 2018 to yield what they

coin a deep potential. In mainstream machine learning literature, the word deep corresponds to a large number of fitting parameters in a network, frequently inside the hidden layers of a network, as opposed to shallow networks with few fitting parameters. The default architecture of deep potential uses five hidden layers with a decreasing number of nodes (240, 120, 60, 30, and 10). Instead of using a symmetry function like Behler-Parinello, deep potential expresses cartesian coordinates of a configuration in local reference frames of each atom.

The main concern with the deep potential method is that a single neural network must be trained for each new configuration and that this network does not explicitly support different atom types. This means two things: (1) It is not possible to train a deep potential for one molten salt mixture and then to transfer it to a slightly different one (FLi to Flibe for example); (2) The neural network is trained on the average distribution of atomic species within a cutoff distance. If on average a NaCl system has 10 Na and 10 Cl atoms as closest neighbors to each atom, then the deep potential will have optimized the weights for the first 10 input nodes for Cl and the last 10 for Na, as they simply sort the neighbor by atom type. If an atom has by chance 8 Na and 12 Cl closest neighbors, the weights for input nodes 9-10 would treat Cl atoms like Na atoms.

Surprisingly, this systematic error has not been more thoroughly discussed by the authors in this or their two influential follow-up papers DeepMD-Kit[110] and DP-Gen[111], which both use this same approach. For users of deep potential, it is especially important to highlight the compounding effect of this setup for complicated molten salt mixtures. With an increasing number of atomic species in the system, the fluctuations in neighborhoods will prohibit accurate energy assessment for individual atoms, while providing decent predictions on average.

The problem of generalizability of the original Behler-Parinello[98] approach was already addressed in 2017 by Smith et al. in their highly influential paper that introduced the ANI-1 network for biochemical simulation.[112] Here, the authors used heavily modified symmetry functions that support encoding of recognizable features in the molecular representations and differentiation between atomic species. Most notably, they train neural networks for atomic species, instead of using a single network for all atoms. While this increases the generalizability of the models, which were trained on small molecules but tested with good accuracy on a larger test set, these networks need to be re-trained simultaneously whenever a new atomic species is added to the data set. The first ANI-1 model supported only 4 atom types (H, O, C, N), the ANI-2 model[113] expanded this to 7 atom types (S, F, Cl). While the ANI-1x dataset[114] is available, the ANI-2 dataset has not been made publicly available. The authors report empirical evidence "that discriminating between atomic numbers allows for training to much lower error on diverse multi-molecule training sets and permits better transferability"[114].

The molten salt community has access to the optimized symmetry functions of Smith et al. through the *Properties from Artificial Neural Network Architectures* (PANNA)[115] software suite that was developed to support the training of Behler-Parinello models. Similar to DeepMD-kit, PANNA integrates with the popular large-scale atomistic/molecular massively parallel simulator (LAMPPS)[116] MD package.

A popular alternative to training custom neural networks for molten salt simulations is fitting of a Gaussian Approximation Potential (GAP) as introduced by Bartok et al.[99] This method uses ab initio configurations to fit a gaussian potential for each atom in the configuration. While these potentials have hyperparameters that can be optimized, the actual fitting of the potentials is fully deterministic opposed to the stochastic processes used in the training of Behler-Parinello ANNs. While

GAP does not fully qualify therefore as a machine learning model, the current literature often describes the fitting of the potentials with the same vocabulary (hyper parameter tuning, test/validation/training datasets, etc). Contrary to empirical potentials discussed earlier, GAP is theoretically able to model any complex potential energy landscape, given a sufficiently large dataset of ab initio configurations and energies for fitting.

*First proofs of principle—review of applications of neural networks to molten salts.* The following overview of recent publications will use either GAP or ANN based methods. All of them follow the same strategy of first running short and small AIMD simulations, fitting a potential specific to their current salt mixture, and then running MD with that potential. Due to limited integrations, all simulations listed here use the LAMPPS package. This strategy overcomes the prohibitive cost of running long AIMD for large systems and holds the promise of higher accuracy than empirical potentials with their limited expressibility.

In 2020, Tovey et al. investigated NaCl with the GAP potential fitted on a AIMD dataset of 1000 configurations, for a system of 64 ions for 14 ps[117]. MD in LAMPPS was afterward run for up to 10 thousand ions for 30 ns. The resulting PRDF agrees very well with data from beam line experiments. Sivaraman et al. trained a GAP model on AIMD configurations from LiCl simulations and reports good alignment between resulting MD and AIMD simulations[118]. Nguyen et al. investigated actinide molten salts by training a GAP on AIMD configurations and showed good agreement between experiments and simulations of $ThCl_4$–NaCl and $UCl_3$–NaCl[119].

Li et al. fitted a Behler-Parinello model with PANNA to AIMD data of NaCl and showed good agreement between the resulting MD trajectory and the AIMD data[120]. Lam et al. also used the Behler-Parinello approach with PANNA to train an ANN for LiF and Flibe[121]. For LiF the mean average error (MAE) of the energy was less than 2.1 and 2.3 meV/atom for liquid and solid LiF respectively which is near the precision of the DFT calculations they used to generate the reference data. For Flibe the MAE in energy was 2.10 meV and a MAE in force of 0.084 eV/Å which is the same order of error inherent in DFT calculations. In a calculation of the ionic diffusion coefficients of Be, F, and Li, all were within ±10–15% error from the DFT calculations which are within the uncertainty of typical experiments.

The Y. Lu lab used the DeepMD-kit to train a deep potential for $ZnCl_2$ on AIMD data with 108 atoms simulated for 30 ps[122]. Larger MD with 1980 atoms and 100 ps duration was compared with a PIM model that simulated 768 atoms for 500 ps. Extracted properties aligned very well between all models, and the authors suggested a deeper comparison between ML and PIM accuracies. In a follow-up work by the Lu lab, the potential was extended to $ZnCL_2$ mixtures. The resulting ML trajectories showed reasonable agreement within 26% of experiment values for thermal conductivity using the RNEMD method, within 6.6% for specific heat capacity, and within 4.2% for density. These larger uncertainties might well be related to the previously discussed problem of deep potential only training a single ANN. For increasingly complex mixtures, the assignment of atom types to correct input nodes becomes error-prone and should be carefully evaluated in such studies.

The G. Lu lab has embraced the deep potential method and published a series of experiments between 2020 and 2022, first to simulate $MgCl_2$[123] and then $MgCl_2$-KCl[124]. Their strategy is always to run a short AIMD simulation of <100 atoms, use DeepMD-Kit to fit a deep potential, and then follow up with a longer MD simulation using the trained potential. They report consistently good agreement with experimental values for similar studies investigating LiCl-KCl mixture[125], alkali chlorides[126],

lanthanum chlorides[127], KCl-CaCl$_2$[128], Li$_2$CO$_3$-Na$_2$CO$_3$[129], and SrCl$_2$[130].

Rodrigues et al. used DeepMD-Kit to fit existing AIMD data for LiF and Flibe to a deep potential before conducting MD that resulted in good agreement with the AIMD data[131]. Li et al. investigate the interactions of Uranium in NaCL[132]. They found that a deep potential model trained on AIMD data outperformed classical PIM models. Lee et al. showed also that FLiNaK can be simulated with higher accuracy using a deep potential model than a RIM model trained on AIMD data[133]. It will be interesting to see if additional reports will substantiate the evidence for deep potentials being superior to the current state-of-the-art empirical potentials and supersede them in the near future.

*Evaluating the status quo—how far can current neural networks go?.* This comprehensive review of machine learning methods for molten salt simulations clearly shows one shortcoming of the applied methodology. None of these papers has reused a model fitted by other groups. Compared with PIM parameters that allow potential reuse by other groups, this breed of ANNs or GAPs always requires new potential fitting before investigating a novel salt mixture. The core problem is not the available algorithms, as demonstrated by the extensible ANI/ANI2 networks that build off a modified Behler-Parinello approach. It is rather that the recent studies are conducted by users and not developers of ML software. Promising new ML models, like graph neural networks such as the SE(3)-transformer[134] are likely going to outperform the current deep potential and Behler-Parinello methods as soon as they become easily trainable and integrated with favorite simulations tools. Especially for more complex mixtures, it will be necessary to make the shift to more extensible architectures, as current approaches suffer too much from the limited expressibility of atom or isotope types. In general, it is worthwhile to remember that machine learning can be optimized through three different approaches: (1) increasing datasets, (2) improved training strategies, (3) better network architectures. While some recent papers try to improve accuracies through increased datasets[120], others apply strategies such as active learning[135] to reduce the necessary number of training data.

It would be beneficial if a shared database of AIMD configurations of multiple salts would be made available (similar to the ANI dataset), ideally in conjecture with experimentally collected thermophysical properties to create community benchmarks.

As these first studies prove the value of machine learning for replacing ab-initio MD methods and empirical force fields, the current breed of applications only scratches the surface of what is possible. Deep learning does not need to be limited to a specific set of atom species, but novel network architectures, such as graph-based neural networks like the SE(3)-transformer[134], could potentially generalize over the entire periodic table of elements and the various isotopes encountered in molten salt reactor simulations. Training of such a general-purpose machine learning model would however require a more community-oriented sharing of training data through open databases and reproducible benchmarks.

Once a more general neural network is available, integration with high-performance MD codes, such as OpenMM[136] or Gromacs[137], could increase the usage further than PANNA or DeepMD-kit's current scopes. The limited support of machine learning potential in LAMMPS is currently restricted to Tensor-flow implementations, the popular PyTorch library however can more easily be integrated with OpenMM. Coupled with the right simulation tools, accurate potentials could be used not only to investigate defined salt mixtures but rather to propose ideal mixtures for desired properties. In an unsupervised optimization process, molten salt constituents and ratios could be developed that outperform known mixtures and result in safer and more efficient reactors and solar systems.

**Summary of potentials**. This section shows the functional forms of all commonly used potentials in molten salt simulations in order of first publication.

*Born-Huggins-Mayer (BHM)[10].*

$$\Phi(r) = -\alpha \frac{e^2}{r} - \frac{C}{r^6} - \frac{D}{r^8} + \beta_{ij} b e^{(\sigma_j + \sigma_i - r_{ij})/\rho} \tag{25}$$

The *Born-Huggins-Mayer* (BHM)[10] model which was developed in 1933 is a pair-potential for akali halides. $\alpha$ is the Madelung constant for the associated ionic crystal. $C$, $D$ are van der Wall constants calculated by Mayer[138]. $\beta_{ij}$ is the Pauling factor[139]. $b$ is an arbitrarily chosen factor. $\sigma_i, \sigma_j$ are the radii of ions. $r_{ij}$ is the separation distance of the ions. $\rho$ is empirically determined.

*Shell Model[13].*

$$\Phi(x, d) = V_e + V_{sr} + V_{se} = \left( -pE - \frac{1}{2}Cp^2 \right) \\ + \frac{1}{2}Ax^2 + \left( \frac{1}{2}k_+ d_+^2 + \frac{1}{2}k_- d_-^2 \right) \tag{26}$$

$$p = e(x_+ - x_-) - n_- e d_- - n_+ e d_+ + D(x_+ + d_+ - x_- - d_-) \tag{27}$$

$$C = 4\pi \frac{N}{3} \tag{28}$$

$$x = x_+ + d_+ - x_- - A d_- = \frac{e^{-\frac{R_0}{\rho}}(R_0 - 2\rho)}{R_0 \rho^2} \tag{29}$$

The Shell Model[13] which was developed in 1958 represents an ion composed of a core and shell of opposing charge. The pair potential consists of three terms: electrostatic, short-range, and self-energy interactions which depend on four generalized coordinates. $p$ is the polarization, $x_+, x_-$ is the distance of the core center to the respective lattice site, $d_+, d_-$ is the distance of the core site to the shell center, $e$ is the elementary charge, $E$ is the macroscopic electric field, $P = N_p$ is the definition of polarization where $n_+$, $n_-$ is the number of electrons, $N$ is the number of ions per unit volume, $R_0$ is the lattice separation, and $\rho$ is the same as in the BHM model. $D$ is the exchange charge polarization coefficient.

*BHM-Tusi-Fumi (BHMTF)[11,15].*

$$\Phi(r) = \frac{Z_i Z_j e^2}{r} + B_{ij} e^{-\alpha_{ij} r} - \frac{C_{ij}}{r^6} - \frac{D_{ij}}{r^8} \tag{30}$$

$$B_{ij} = c_{ij} b e^{\frac{(\sigma_i + \sigma_j)}{\rho}} \tag{31}$$

$$c_{ij} = 1 + \frac{Z_i}{n_i} + \frac{Z_j}{n_j} \tag{32}$$

$$\alpha_{ij} = \frac{1}{\rho} \tag{33}$$

BHM-Tusi-Fumi (BHMTF)[16] model which was developed in 1964 where the effective pair potential differs from the original BHM model by allowing $\rho$ to vary from salt to salt determined semi-empirically and using the effective charge $Z_i Z_j$ of ion pairs.

$n_i, n_j$ is the number of electrons in the outer shell. All the other parameters are the same as the BHM model[11].

*Tang-Toennies damping function for dispersion[20].*

$$\Phi(r) = V_{scf} - \sum_{n>3}^{\infty} \frac{f_{2n}(r) C_{2n}}{r^{2n}} \tag{34}$$

$$f_{2n}(r) = 1 - \left( \sum_{k=0}^{2n} \frac{(bR)^k}{k!} \right) e^{-bR} \tag{35}$$

The damping function developed in 1984 represents a generalized way to damp the polarization dispersion energy. The accuracy was verified on several ab-initio calculations.

*Madden-Wilson Model[82].*

$$\Phi(r) = \sum_{j \neq i}^{N} B_{ij} e^{-\alpha_{ij} r} - \frac{C_{ij}}{r^6} - \frac{D_{ij}}{r^8} + Z_i \left( \sum_{\nu=1}^{2} \frac{q_{\nu i}}{r_\nu} + \frac{Z_j}{r} \right) + V_{\text{dipole}} \tag{36}$$

$$V_{\text{dipole}} \approx \mu_i \nabla_{r_i}^2 \left( - \left( \sum_{\nu=1}^{2} \frac{q_{\nu i}}{r_\nu} + \frac{Z_j}{r} \right) \right); \; \mu_i = q_i d_i \tag{37}$$

The Madden-Wilson model developed in 1993 includes the original BHM terms with a change of the first coulombic charge to include the variable charge, $\nu$, of the ion and a dipole potential. The dipoles are approximated with the form given, where $d_i$ is the rod length of the dipoles.

*Compressible Ion Model[83].*

$$\Phi(r) = u_{CI}(r, \delta_i) + F(\delta_i) + u^{--}(r) + \frac{Z_i Z_j}{r} + \left( 1 - f_6^{ij}(r) \right) \frac{C_{ij}}{r^6} - \left( 1 - f_8^{ij}(r) \right) \frac{D_{ij}}{r^8} \tag{38}$$

This pair potential, developed in 1996, is for oxide-type salts. It is the BHM potential with dispersion Tang-Toennies damping functions and modification of the repulsion term. The ionic radii are allowed to vary and described through the relation $\sigma_i = \bar{\sigma}_i + \delta_i$ where $\bar{\sigma}$ is the average ionic radii and $\delta_i$ describes instantaneous changes based on the environment. The first two terms account for the repulsion term in BHM while the third term, $u^{--}$, accounts for a frozen oxide-oxide interaction.

*Polarizable Ion Model—induced dipole and quadrupole[88].*

$$\Phi(r, \mu, \theta) = \sum_{i,j \in \text{all ions}} \left[ \left( Q^i \mu_\alpha^j - Q^j \mu_\alpha^i \right) T_\alpha^{(1)} + \left( \frac{Q^i \theta_{\alpha\beta}^j}{3} + \frac{\theta_{\alpha\beta}^i Q^j}{3} - \mu_\alpha^i \mu_\beta^j \right) T_{\alpha\beta}^{(2)} \right.$$
$$+ \left( \frac{\mu_\alpha^i \theta_{\beta\gamma}^j}{3} + \frac{\theta_{\alpha\beta}^i \mu_\gamma^j}{3} \right) T_{\alpha\beta\gamma}^{(3)} + \frac{\theta_{\alpha\beta}^i \theta_{\gamma\delta}^j}{9} T_{\alpha\beta\gamma\delta}^{(4)} \right]$$
$$+ \sum_{i \in \text{anions}, j \in \text{cations}} \left[ Q^j \mu_\alpha^i T_\alpha^{(1)} f_{ij}(r) + \frac{\theta_{\alpha\beta}^i Q^j}{3} T_{\alpha\beta}^{(2)} g_{ij}(r) \right]$$
$$+ \sum_{i \in \text{anions}} [k_1 \mu^{i^2} + k_2 \mu_\alpha^i \theta_{\alpha\beta}^i \mu_\beta^i + k_3 \theta_{\alpha\beta}^i \theta_{\alpha\beta}^i + k_4 \mu^{i^4}] \tag{39}$$

This potential, developed in 2002, depends on the dipole moment, μ, the quadrupole moment, θ, and Q, the formal charge of the ion. The $T_{\alpha\beta\gamma}^i \ldots$ is the interaction tensors, while $k_i$ are proportional to polarizabilities.

*Aspherical Ion Model[25,28].*

$$\Phi(\{R^N\}) = \sum_{i<j} A e^{-(a(r_{ij} - \sigma^i(\{R^N\}) - \sigma^j(\{R^N\})))} + \sum_i u_{\text{self}}(\sigma(\{R^N\}) +$$
$$+ \sum_{i<j} \left( q^i \mu_\alpha^j (1 - c_{ij} g(r_{ij})) - q^j \mu_\alpha^i (1 - c_{ji} g(r_{ij})) T_\alpha^{(1)} - \mu_\alpha^i \mu_\beta^j T_{\alpha\beta}^{(2)} + \sum_i k^i |\mu^i|^2 \right) \tag{40}$$

This model, developed partially in 1998 and finished in 2006 combines the Compressible Ion Model and the Polarizable model.

*Polarizable Ion Model—induce dipole (PIM)[45].*

$$\Phi(r) = \sum_{j<i}^{N} \left( B_{ij} e^{-\alpha_{ij} r} - \left( 1 - f_6^{ij}(r) \right) \frac{C_{ij}}{r^6} - \left( 1 - f_8^{ij}(r) \right) \frac{D_{ij}}{r^8} + \frac{q_i q_j}{r} \right.$$
$$+ \sum_{i<j} \left( q^i \mu_\alpha^j (1 - c_{ij} g(r_{ij})) - q^j \mu_\alpha^i \left( 1 - c_{ji} g(r_{ij}) \right) T_\alpha^{(1)} - \mu_\alpha^i \mu_\beta^j T_{\alpha\beta}^{(2)} + \sum_i k^i |\mu^i|^2 \right) \tag{41}$$

The model, developed in 2008 is reduced to dipole contribution plus the BHMTF model. The dipole moments are calculated self consistently.

*Behler-Parinello Neural Network (BPNN)[98].*

$$E_i = f_a^2 \left[ w_{01}^2 + \sum_{j=1}^{3} w_{j1}^2 f_a^1 \left( w_{0j}^1 + \sum_{\mu=1}^{2} w_{\mu j}^1 G_i^\mu \right) \right], \tag{42}$$

where $w_{ij}^k$ is the weight parameter connecting node $j$ in layer $k$ with node $i$ in layer $k - 1$, and $w_{0j}^k$ is a bias weight that is used as an adjustable offset for the activation functions $f_a^k$: hyperbolic tangent for hidden layers and liner function for output. Weights were initially randomly chosen but error functions can be minimized to obtain correct weights. $G_i^\mu$ is a symmetry function of the cartesian coordinates $R_i^\alpha$ of the local environment of the $i$ atom. The locality is determined by the function

$$f_c(R_{ij}) = \begin{cases} \frac{1}{2} \left[ \cos \left( \frac{\pi R_{ij}}{R_c} \right) + 1 \right] & \text{for } R_{ij} \leq R_c \\ 0 & \text{for } R_{ij} > R_c \end{cases} \tag{43}$$

where $R_c$ is some predetermined cutoff radius. The radial symmetry function is

$$G_i^1 = \sum_{j \neq i}^{\text{all}} e^{-\eta (R_{ij} - R_s)^2} f_c(r_{ij}), \tag{44}$$

where $\eta, R_s$ are parameters. The angular symmetry function is

$$G_i^2 = 2^{1-\zeta} \sum_{j,k \neq i}^{\text{all}} \left( 1 + \lambda \cos \theta_{ijk} \right)^\zeta e^{-\eta \left( R_{ij}^2 + R_{ik}^2 + R_{jk}^2 \right)} f_c(R_{ij}) f_c(R_{ik}) f_c(R_{jk}), \tag{45}$$

where $\xi, \lambda \in \{-1, 1\}$ are new parameters and $\cos \theta_{ijk} = \frac{\mathbf{R}_{ij} \cdot \mathbf{R}_{jk}}{R_{ij} R_{jk}}$ with $i$ being the central atom and $\mathbf{R}_{\alpha\beta} = \mathbf{R}_\alpha - \mathbf{R}_\beta$

*Deep potential[108].*

$$E_i = f \left[ w_i^\alpha, I_i \right], \tag{46}$$

where $f, w$ comes from a fully-connected feedforward neural network. $I_i$ is the input vector of the function: it takes in the positions of all the atoms (in cartesian or some polar-like coordinate system $\left\{ \frac{1}{r}, \cos \theta, \cos \phi, \sin \phi \right\}$) centered on the $i$ atom with a cutoff of the nearest $N_c$ neighbor atoms determined from the max number of atoms within the cutoff radius of $R_c$. The weights are constrained to be the same for atom-types $\alpha$. The ordering of the inputs into $I_i$ at the first level is atom-type $\alpha$ and at the second level ascending distance from the $i$ atom.

*Deep Potential Molecular Dynamics (DeepMD)*[109].

$$E_i = f\left[w_i^\alpha, \mathbf{D}_{ij}\right], \tag{47}$$

where $E_i$ comes from Deep Potential except for $I_i$ is called $D_{ij}$:

$$\mathbf{D}_{ij} = \begin{cases} \left\{\frac{1}{R_{ij}}, \frac{\mathbf{x}_{ij}^\alpha}{R_{ij}}\right\} \\ \frac{1}{R_{ij}} \end{cases}, \tag{48}$$

where $R_{ij}$ is the separation between atoms $i,j$ and $\mathbf{x}_{ij}^\alpha$ are the cartesian components of the separation vector $\mathbf{R}_{ij}$ and $\mathbf{D}$ is ambiguous in which situation which case should be used. The parameters are determined by the Adams method, see the associated paper for more details. The parameters are from the fitting functions which are of the similar form described in BPNN.

*Gaussian Approximation Potentials (GAP)*[99].

$$E = \sum_i^{atoms} \epsilon\left(\left\{\mathbf{r}_{ij}\right\}\right), \tag{49}$$

represents the total energy of the system where $\mathbf{r}_{ij}$ is the separation vector and $\epsilon$ is the atomic energy function. GAP uses a localized version of the above equation replacing the separation vector with the truncated spectrum space: $\epsilon\left(\left\{\mathbf{r}_{ij}\right\}\right) \to \epsilon(\mathbf{b})$. The spectrum space is constructed from the local atomic density of each atom $i$ with a cutoff radius determining the locality and projected into 4D spherical harmonics which encapsulates the 3D spherical coordinates. The projections form a basis set with the Clebsch-Gordon coefficients to determine the intensity of each projection.

$$a_{m'm}^j = \left\langle U_{m'm}^j, |, \rho \right\rangle, \tag{50}$$

$$\mathbf{b}_i = \sum_{\substack{(m_1', m_1 = -j_1), \\ (m_2', m_2 = -j_2), \\ (m', m = -j)}}^{j_1 j_2 j} a_{m'm}^j C_{j_1 m_1 j_2 m_2}^{jm} \times C_{j_1 m_1' j_2 m_2'}^{jm'} a_{m_1' m_1}^{j_1} a_{m_2' m_2}^{j_2}, \tag{51}$$

where the index $i$ is dropped in the equations above and the following definitions: $U_{m'm}^j$ are the Wigner matrices (4D spherical harmonics), $\rho$ is the local atomic density of the $i$ atom, $C_{j_1 m_1 j_2 m_2}^{jm}$ are the Clebsch-Gordon coefficients, and the total angular momentum is truncated, $j, j_1, j_2 \leq J_{max}$, to the associated atomic neighborhood.

The atomic energy function is approximated as a series of Gaussians:

$$\epsilon(\mathbf{b}) = \sum_n \alpha_n e^{-\frac{1}{2}\sum_i \left(\frac{b_l - b_{n,l}}{\theta_l}\right)^2} \equiv \sum_n \alpha_n G(\mathbf{b}, \mathbf{b}_n), \tag{52}$$

where $n,l$ run over the reference configuration and spectrum components, $\alpha_n$ is the fitting parameter, and $\theta_l$ is a hyper parameter. The function is fit by least-squares fitting using the covariance matrix:

$$C_{nn'} = \delta^2 G(\mathbf{b}, \mathbf{b}') + \sigma^2 \mathbb{II}, \tag{53}$$

$$\{\alpha_n\} \equiv \boldsymbol{\alpha} = \mathbf{C}^{-1}\mathbf{y}, \tag{54}$$

where $\delta, \sigma$ are hyperparameters and $\mathbf{y}$ is the set of reference energies.

**Comparing accuracy of models based on derived properties**. Sections "Early simulations 1933–1990, Progression of DFT-based methods 1990–2021, Machine Learning Era 2020—

**Table 1 LiCl bond-lengths from experiments and simulations.**

| Method | First peak position (Å) | Temperature (K) | Citation |
|---|---|---|---|
| AIMD | 2.41 | 873 | [147] |
| | 2.36 | 958 | [148] |
| | 2.00 | 1173 | [147] |
| Experiment | 2.40 | 923 | [149] |
| | 2.45 | 933 | [150] |
| | 2.29 | 958 | [149] |
| RIM | 2.22 | 958 | [148] |
| | 2.21 | 1100 | [151] |
| | 2.25 | 1100 | [7] |
| PIM | 2.30 | 958 | [148] |
| GAP | 2.30 | 900 | [118] |
| DeePMD | 2.37 | 900 | [126] |
| | 2.39 | 1000 | |
| | 2.41 | 1100 | |
| | 2.35 | 1200 | |

Comparison of predictions from simulation methods with experiment for LiCl at different temperatures.

ongoing" have summarized the various computational methods deployed to simulate molten salts: ab-initio molecular dynamics (AIMD), rigid ion model (RIM), polarizable ion model (PIM), Gaussian Approximation Potential (GAP), and deep potential learning methods (DeePMD). A comprehensive review of all computational potentials reported has been provided in Section 3.4. Here, we use metanalysis to provide a comparison between the different methods.

Comparing the accuracy of models quantitatively is impractical due to the lack of consistency in reporting properties across the literature and the nature of the derived properties. For thermophysical properties, the data is primarily in graphical form to capture the relationship between temperature and that property, but for structural properties, some numerical values, such as coordination number and position of local minimums and maximums, are reported in the literature.

We compare the methods using the first peak position of the PDF of Li-Cl ions as shown in Table 1. This value represents the average bond length of the Li-Cl ion and presents a reasonable way to compare the accuracy of models. AIMD and Experiment agree well, with AIMD capturing the descending behavior with respect to increasing temperature. PIM, GAP, and DeePMD agree reasonably well with AIMD (the value of GAP was not directly reported but extracted from a graph by visual approximation). RIM does not capture the temperature dependent behavior, which underlines the importance of polarization effects as widely described in the literature.

Another frequently simulated salt is Flibe, for the first peak position of the partial radial distribution function we find good agreement across PIM[80], RIM[140], AIMD[35,44], X-ray diffraction experiment[141], and ML methods[121,131]. Only the modified RIM shows disagreement of 0.1 Å for the fluoride-fluoride pair (see Supplementary Data 1—Table S1).

Multiple thermophysical properties of Flibe have been reported across different methods. For density: RIM agrees to 0.5% with experiment[142]; PIM didn't report density but demonstrated good agreement with experimental molar volumes[80]; similarly, the DFT methods reported agreement with experimental equilibrium volumes[143,144] by (9–18)% with better agreement using PBE with dispersion corrections (vdw-DF[145]) with 2% inaccuracy[146]; ANI-1 underpredicted density by 6%, potentially because this study didn't use a training dataset that corrected for dispersion[121]; DeePMD underpredicted density by 14% compared to experiment[131].

For diffusion, it is difficult to define an accurate benchmark as there have not been many experimental measurements and the accuracy of AIMD is limited due to short time scales. Compared to AIMD, ANI-1 and DeePMD show 10–15% agreement while PIM shows agreement to within 30%. There is no report for RIM on diffusion coefficients, but we don't expect it to be better based on the viscosity and electrical conductivity data.

For Electrical conductivity: RIM is close to experimental values for higher temperatures but deviates much further for lower temperatures, PIM shows better agreement with experimental data, and DeePMD appears to do worse than PIM but not by much. Thermal conductivity was overestimated by 20% with PIM compared to accepted experimental results. DPMD finds better agreement than PIM. ANI-1 didn't report thermal or electrical conductivities; however, we expect a similar or even better performance to DeePMD.

For viscosity, DeePMD performs best followed by PIM then by RIM, other methods did not report viscosities. Lam et al. compared computational times of PIM, AIMD, and AN1-1 and found that the computational time in ascending order was PIM, ANI-1, then AIMD demonstrating that PIM is faster than ANI-1[121].

It is unclear what may definitively be said about the best model due to inconsistency in reported values across all thermophysical properties. The ML methods show promise with ANI-1 and GAP demonstrating the greatest accuracy but are suspected to suffer from transferability across compositions where PIM may out-perform due to its parameters originating from a larger composition space[91].

## Conclusions and outlook

Accurate prediction of thermophysical properties of molten salts can have an immeasurable impact on our society as new reactor and solar power systems are being developed. Throughout the last 80 years, breakthroughs in theory, computational power, and experiments have substantially advanced our ability to extract the necessary properties from simulations. Today, we witness the advent of machine learning in molten salt simulations and foresee unprecedented improvements in our abilities to design and use new molten salt mixtures. Machine learning models present a valuable midground between the accuracy and efficiency of classical force fields and ab-inito calculations. Thermophysical properties derived from extended simulations will be able to fit more accurate computational fluid dynamic models and help in designing superior molten salt systems.

To support the development of next-generation machine learning methods we identify the following concrete needs: (1) Existing data from DFT calculations should be made freely accessible and transparent to enable data science groups to train novel models; (2) high-performance integrations of machine learning forces into existing molecular dynamics toolkits should be extended beyond the current LAMMPS integrations; (3) reusability of machine learning models should be increased by proper sharing and documentation of models; (4) a set of experimental benchmarks should be defined for a representable set of molten salts to allow for reproducible assessment of the quality of predicted thermophysical properties; (5) an open library for the analysis of trajectories and extraction of thermophysical properties should be made available to render results amongst studies more comparable.

Additional opportunities that may lead to improved machine learning models include the following: (1) Design of custom machine learning architectures for molten salts that incorporate inductive biases, which will require close collaboration between chemists and computer scientists; (2) introduction of superior active learning techniques to develop optimal training strategies; (3)

development of architectures that can generalize beyond a few atom types, ideally to the space of all relevant molten salts and solvents.

The grand challenge of molten salt simulations is optimizing salt mixtures for desired thermophysical properties. Current machine learning models are still limited to predictions for mixtures that were used to train them and small variations in experimental conditions, such as temperature. We expect that better datasets, training strategies, and architectures will soon overcome these limitations.

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

## Acknowledgements

The authors gratefully acknowledge financial support for undergraduate assistantships from Brigham Young University College of Physics and Mathematical Sciences. MV was partially supported by a WAESO grant under number F2020ur0009.

## Author contributions

T.P., M.M.V., and D.D.C. designed the study. T.P. and M.M.V. reviewed the majority of papers in Sections "Early simulations 1933-1990" and "Progression of DFT-based methods 1990–2021", P.S. reviewed the extraction of thermophysical properties. M.M.V., P.S., and D.D.C. reviewed the papers about machine learning. The manuscript was written through the contributions of all authors. All authors have approved the final version of the manuscript.

## Competing interests

The authors declare no competing interests.
