## [Peer Review File · Communications Chemistry]

Reviewers' comments:

Reviewer #1 (Remarks to the Author):

This review paper provides some recent progress in application of machine learning in studying thermodynamic properties of molten salts. The reviewer has the opinion that, the relevant machine learning part is not fully comprehensive, while the classical methods and/or background calculations have been thoroughly discussed and presented. The reviewer would suggest adding more discussion and presentation for machine learning part, which would be more beneficial for the readers who want to gain more understanding and knowledge about the state-of-the-art techniques. Some challenges for machine learning need to be more specific and expanded, for instance, how to expand the neural network potential for new mixtures? How to find ideal components of new mixtures with desired thermodynamic properties using machine learning approaches? In addition, some recent papers in machine learning for studying molten salts have not been included in the references: ACS Appl. Mater. Interfaces 2021, 13, 55367–55379.

Reviewer #2 (Remarks to the Author):

Porter and co-workers offer a limited review of computational models for the thermochemical properties in molten salts, with focus on the more recent machine learning approaches. Overall, the review may be quite useful as a reference of various models that have been employed in the literature, but it is mostly skin-deep. What I mean is that they basically list some papers on the subject and provide a brief description of what each paper is about, but they do not really discuss the advantages or shortcomings of these models. I think it would be very helpful to provide some guidance as to which model would be best for what property, for example.

The title is also a bit misleading, as it implies that the paper mostly discusses ML advances in this field, and it does, but the ML part is a very limited part of this paper, a couple of pages in total. I suggest the authors modify the title and maybe try and include some discussion regarding recommendation on models.

The authors should do another quick search for relevant references that they omitted, see for example below:

Besmann, T.M. and Schorne-Pinto, J., 2021. Developing Practical Models of Complex Salts for Molten Salt Reactors. *Thermo*, 1(2), pp.168-178.

Nguyen, M.T., et al. 2021. Ab initio molecular dynamics assessment of thermodynamic and transport properties in (K, Li) Cl and (K, Na) Cl molten salt mixtures. *Journal of Molecular Liquids*, 326, p.115262.

Schorne-Pinto, J., et al 2021. Correlational Approach to Predict the Enthalpy of Mixing for Chloride Melt Systems. *ACS omega*.

The manuscript could be accepted after considering the comments and revising accordingly.

Reviewer #3 (Remarks to the Author):

The manuscript by Porter et al. reports a comprehensive review of past experimental research and simulations on molten salt (MS) in the past 70 years. More recently when methods using machine learning (ML) got started a few years ago. The paper is directed by a young, energetic researcher with a somewhat different career path. I am quite impressed by this nonconventional topic of historical importance and the way it is presented. It suffers some drawbacks due to lack of experience but can be fixed. I recommend this paper be accepted by Communication Chemistry after a major revision. I summarize my assessments below.

A. Strong points

1. The paper contains an extensive list of references which is important for a review article. The Introduction is clearly written for the historical account on MS. Obviously, a team of diligent undergraduate students are involved. I consider this to be a very positive endeavor by submitting their collective work to a prestigious journal.
2. The discussion part consists of the of the development and research for MS into two different era: from 1933 to 1990 for experimental work and from 1990 to 2021 for simulation work 2021. This is discussion is very useful and appropriate for a review article especially for the historical account for the early period.
3. The flow chart of Figure 3 is quite comprehensive and novel. It can be improved by a more detailed descriptions in the caption about the items and numbers used in the chart.

B. Weak points

1. The paper is careless written with no page numbers making it difficult for reviewers to refer to.
2. Figure 2 is useful and informative but can improved and better designed. Is there any properties related to mechanical properties or optical properties? For MS, optical properties are very important and may be the only major experimental effort because of the toxicity of almost all MS.
3. Equations should be clearly numbered and consecutively listed for easy understanding. In currently presentation, it is jumping around and very confusing.
4. Table 1 is very important on the summary of MS potential but poorly designed and awkward. It occupies too large amount of precious journal space and should be significantly improved.
5. I consider the period 3 from 2020 to present on ML to be the most important and focal point for this paper. Please expand it to include more details of the current methods and approaches (many different types exist) and the role of the quality of the data used in ML. Can data from accurate simulations be used in ML for complex biological materials? In biomolecular arena, the data used for ML are from many different sources and some of them are of questionable quality or entirely irrelevant.

C. Minor points

1. I prefer the major section such as INTRODUCTION and other subsections to be numbered. This may be due to the Journal policy and specification and the authors are just following the guideline.

2. In figure 1, can the authors extend the dimension of the figure to cover the entire cell of 1024 atoms in Fig. 1(a) and 200 atoms in in Fig. 1(b) for 200 atoms?

3. There is one very recent work on MS simulations which should be cited. ([DOI. org/10.1021/acsomega. 1c02528](https://doi.org/10.1021/acsomega.1c02528)). Apparently, it appeared after the authors submitted their manuscript.

4. There is a typo error on the page containing Figure 3. Should it be Figure 3 rather than Figure 1 on the line above the figure. There could be other typo errors and the authors need to check carefully.

Point-by-Point Response

Advent of Machine Learning for Molten Salt Simulations

Porter et al.

Reviewers' comments:

Reviewer #1 (Remarks to the Author):

This review paper provides some recent progress in application of machine learning in studying thermodynamic properties of molten salts. The reviewer has the opinion that, the relevant machine learning part is not fully comprehensive, while the classical methods and/or background calculations have been thoroughly discussed and presented.

→ We agree with the reviewer and have substantially reworked the machine learning chapter of the paper to provide an in-depth discussion of the current state of the art.

Some challenges for machine learning need to be more specific and expanded, for instance, how to expand the **neural network potential for new mixtures?**

→ This is a critical point, especially given that frequently used neural networks do not work well for complex mixtures. The updated sections 3.3 and 3.5 are now discussing this aspect in detail.

In addition, some recent papers in machine learning for studying molten salts have not been included in the references: ACS Appl. Mater. Interfaces 2021, 13, 55367–55379.

→ We repeated the literature search and found multiple additional references that were now added, including this one.

Reviewer #2 (Remarks to the Author):

Porter and co-workers offer a limited review of computational models for the thermochemical properties in molten salts, with focus on the more recent machine learning approaches.

Overall, the review may be quite useful as a reference of various models that have been employed in the literature, but it is mostly skin-deep. What I mean is that they basically list some papers on the subject and provide a brief description of what each paper is about, but they do **not really discuss the advantages or shortcomings of these models**. I think it would be very helpful to provide some guidance as to which **model would be best for what property**, for example.

→ We added a new section (3.5) to the manuscript that discusses the advantages and shortcomings of the identified models. Within 3.3, we particularly analyze the ability of machine learning models using current architectures to generalize. The updated manuscript should now provide more of the information this reviewer suggested.

The title is also a bit misleading, as it implies that the paper mostly discusses ML advances in this field, and it does, but the ML part is a very limited part of this paper, a couple of pages in total. I suggest the authors modify the title and maybe try and include some discussion regarding recommendation on models.

→ We fully agree with this assessment, but instead of changing the title we fundamentally reworked the machine learning section of the article (3.3). In section 3.5 we are now offering additional discussion on how to use the models.

The authors should do another quick search for relevant references that they omitted, see for example below:

Besmann, T.M. and Schorne-Pinto, J., 2021. Developing Practical Models of Complex Salts for Molten Salt Reactors. Thermo, 1(2), pp.168-178.

Nguyen, M.T., et al. 2021. Ab initio molecular dynamics assessment of thermodynamic and transport properties in (K, Li) Cl and (K, Na) Cl molten salt mixtures. Journal of Molecular Liquids, 326, p.115262.

Schorne-Pinto, J., et al 2021. Correlational Approach to Predict the Enthalpy of Mixing for Chloride Melt Systems. ACS omega.

→ We updated the literature search to take the suggested and additional recent articles into consideration.

The manuscript could be accepted after considering the comments and revising accordingly.

Reviewer #3 (Remarks to the Author):

The manuscript by Porter et al. reports a comprehensive review of past experimental research and simulations on molten salt (MS) in the past 70 years. More recently when methods using machine learning (ML) got started a few years ago. The paper is directed by a young, energetic researcher with a somewhat different career path. I am quite impressed by this nonconventional topic of historical importance and the way it is presented. It suffers some drawbacks due to lack of experience but can be fixed. I recommend this paper be accepted by Communication Chemistry after a major revision. I summarize my assessments below.

→ We appreciate these comments, for a team of undergraduate authors this is highly encouraging.

A. Strong points

1. The paper contains an extensive list of references which is important for a review article. The Introduction is clearly written for the historical account on MS. Obviously, a team of diligent undergraduate students are involved. I consider this to be a very positive endeavor by submitting their collective work to a prestigious journal.
2. The discussion part consists of the of the development and research for MS into two different era: from 1933 to 1990 for experimental work and from 1990 to 2021 for simulation work 2021. This is discussion is very useful and appropriate for a review article especially for the historical account for the early period.
- 3. The flow chart of Figure 3 is quite comprehensive and novel. It can be improved by a more detailed descriptions in the caption about the items and numbers used in the chart.**

→ We updated Figure 3 to contain additional information and provided an updated legend as suggested.

B. Weak points

- 1. The paper is careless written with no page numbers making it difficult for reviewers to refer to.**

→ We misunderstood the author guide lines and restructured the document for improved readability (page numbers, section numbers, etc).

- 2. Figure 2 is useful and informative but can improved and better designed. Is there any properties related to mechanical properties or optical properties?**

→ We reformatted figure 2 to improve its readability. We also expanded the properties by adding structural information like coordination number or bond angles. Optical properties are rarely subject of simulation studies, but we included the examples that could be found.

For MS, optical properties are very important and may be the only major experimental effort because of the toxicity of almost all MS.

→ We added the following paragraph to section 3.2:

Optical properties such as vibrational spectra are more accessible experimentally than computationally and only a small number of DFT studies investigated them^{34, 45}. In 2021, Khagendra et al. reported mechanical properties of Flibe using DFT suggesting that these properties could validate models⁴⁵. However, as these properties are not consistently presented throughout the literature there is currently limited development in this domain.

- 3. Equations should be clearly numbered and consecutively listed for easy understanding. In currently presentation, it is jumping around and very confusing.**

→ We agree with this assessment and have added equation numbers throughout the document.

4. Table 1 is very important on the summary of MS potential but poorly designed and awkward. It occupies too large amount of precious journal space and should be significantly improved.

→ After careful consideration we decided to recast the table as a standalone section, which is now found in section 3.4 as a summary of all the model introduced in 3.1-3.3. This formatting choice appears to increase readability.

5. I consider the period 3 from 2020 to present on ML to be the most important and focal point for this paper. Please expand it to include more details of the current methods and approaches (many different types exist) and the role of the quality of the data used in ML. Can data from accurate simulations be used in ML for complex biological materials? In biomolecular arena, the data used for ML are from many different sources and some of them are of questionable quality or entirely irrelevant.

→ We agree and substantially rewrote section 3.3 to address these concerns. Also, section 3.5 was added to discuss different potentials more thoroughly.

C. Minor points

1. I prefer the major section such as INTRODUCTION and other subsections to be numbered. This may be due to the Journal policy and specification and the authors are just following the guideline.

→ We changed this according to the reviewer's suggestion.

2. In figure 1, can the authors extend the dimension of the figure to cover the entire cell of 1024 atoms in Fig. 1(a) and 200 atoms in in Fig. 1(b) for 200 atoms?

→ We redid figure 1 to remove the unnecessary white space over panel 1 b).

3. There is one very recent work on MS simulations which should be cited. (DOI. [org/10.1021/acsomega.1c02528](https://doi.org/10.1021/acsomega.1c02528)). Apparently, it appeared after the authors submitted their manuscript.

→ This as well as other recent manuscripts have been added to the review.

4. There is a typo error on the page containing Figure 3. Should it be Figure 3 rather than Figure 1 on the line above the figure. There could be other typo errors and the authors need to check carefully.

→ This and other typos have been removed after careful re-read of the manuscript.

REVIEWERS' COMMENTS:

Reviewer #1 (Remarks to the Author):

The review paper is recommended for publication now.

Reviewer #3 (Remarks to the Author):

I am fully satisfied with authors' response and modification. The paper should be accepted.